

# Gene expression analysis of potato drought-responsive genes under drought stress in potato (*Solanum tuberosum* L.) cultivars

Sadettin Çelik

Genç Vocational School, Forestry Department, Bingol University, Bingol, Turkey

## ABSTRACT

The potato (*Solanum tuberosum* L.), an important field crop consumed extensively worldwide, is adversely affected by abiotic stress factors especially drought. Therefore, it is vital to understand the genetic mechanism under drought stress to decrease loose of yield and quality . This trial aimed to screen drought-responsive gene expressions of potato and determine the drought-tolerant potato cultivar. The trial pattern is a completely randomized block design (CRBD) with four replications under greenhouse conditions. Four cultivars (Brooke, Orwell, Vr808, Shc909) were irrigated with four different water regimes (control and three stress conditions), and the gene expression levels of 10 potato genes were investigated. The stress treatments as follows: Control = 100% field capacity; slight drought = 75% field capacity; moderate drought = 50% field capacity, and severe drought 25% field capacity. To understand the gene expression under drought stress in potato genotypes, RT-qPCR analysis was performed and results showed that the genes most associated with drought tolerance were the *StRD22* gene, *MYB domain transcription factor*, *StERD7*, *Sucrose Synthase (SuSy)*, *ABC Transporter,* and *StDHN1*. The *StHSP100* gene had the lowest genetic expression in all cultivars. Among the cultivars, the Orwell exhibited the highest expression of the *StRD22* gene under drought stress. Overall, the cultivar with the highest gene expression was the Vr808, closely followed by the Brooke cultivar. As a result, it was determined that potato cultivars Orwell, Vr808, and Brooke could be used as parents in breeding programs to develop drought tolerant potato cultivars.

## INTRODUCTION

The potato (*Solanum tuberosum* L.) plant from the *Solanaceae* family has remarkable importance as a food crop worldwide (*Camire, Kubow & Donnelly, 2009*). It is a crucial source of healthy carbohydrates, protein, vitamins, mineral, and dietary fiber (*Beals, 2019*). The plant, the third most produced food after wheat and rice (*Stokstad, 2019*), is highly susceptible to long-term water deficiency during its growth and developmental phase. Drought caused by global climate change and extreme temperatures destroys potato-planted areas, consequently limiting potato yield (*Muthoni & Kabira, 2016*). Water

Corresponding author
Sadettin Çelik,
sadettincelik@bingol.edu.tr

stress during this period significantly decreases growth, yield and quality (*Rykaczewska, 2017*). In particular, potato tuber growth and initiation, and bulk periods of growing are highly sensitive to drought stress (*Evers et al., 2010*).

Potato genotypes with shallow root systems have low drought tolerance since the plant cannot reach water deep in the soil (*Zarzyńska, Boguszewska-Mańkowska & Nosalewicz, 2017*). Exposure to drought stress during the developmental stage of the plant, cultivated mainly for its tuber (*Gervais et al., 2021*), leads to reduced leaf area, a diminished photosynthesis rate (*Kutlu & Kinaci, 2010*; *Olgun et al., 2014*; *Çelik, 2023*), poor tuber formation, the shriveling of leaves, reduced tuber yield (*Obidiegwu, 2015*), and a reduced number of tubers per root (*Eiasu, Soundy & Hammes, 2007*). The potato closes its stomata to decrease water loss through transpiration. However, this action causes a decrease in the rate of photosynthesis as the $CO_2$ needed for this process cannot enter through the stomata.

Numerous studies have been carried out to increase the tolerance of the potato plant to biotic and abiotic stress and increase the yield and quality of the unit area in potato plants in recent years is the first aim of many researchers. The large gene pool of the wild form of the potato enables the transmission of biotic and abiotic stress-resistant genes from wild species to cultured species through interspecific hybridization (*Machida-Hirano, 2015*). Marker-assisted selection (MAS) (*Barone, 2004*) and genomic selection (*Meuwissen, Hayes & Goddard, 2001*) increase breeding to develop cultivars resistant to stress factors (*Enciso-Rodriguez et al., 2018*). Various traditional and molecular studies have been conducted to identify the genetic loci responsible for the agricultural features in diploid ($2n = 2x = 24$) potato plants (*Mani & Hannachi, 2015*; *Gebhardt, 2016*; *Anithakumari et al., 2011*; *Anithakumari et al., 2012*). Recent advances in high  high-throughput gene sequencing technologies enable qualitative and quantitative analysis of gene expression and RNA sequence analysis providing high-resolution analysis allows a better understanding of the genetic background of many physiological and agronomic characteristics of field crops (*Wang et al., 2016*; *Prince et al., 2015*; *Gramazio et al., 2016*). Gene expression encompasses the steps from DNA transcription to translation and protein synthesis in ribosomes (*Phillips, 2008*; *Ralston, 2008*).

A great number of molecular studies at the RNA level have been conducted on potato gene activity under environmental stress factors. For insurance, potato *AREB/ABF/ABI5* gene family members were subjected to abscisic acid (ABA) and osmotic stress and the expression of the *StAREB1*, *StAREB2*, and *StAREB4* genes significantly increased in the first six hours, followed by a slight decrease after 24 h (*Liu et al., 2019*). qRT-PCR results indicate that the *AREB/ABF/ABI5* gene members are notably affected under osmotic stress and exhibit a distinct gene expression response to abiotic stress (*Liu et al., 2019*). The *StDHN1*, *StTAS14*, *StERD7*, *StRD22*, and *StHSP100* genes have been identified through real-time PCR-based gene expression analyses to play a role in adaptive stress responses (*Musse et al., 2021*). The overexpression of the *NAC domain* gene *GmNAC06* in soybean plants, specifically in hairy roots, has been observed to increase the plant's salt tolerance significantly. It has been concluded that it plays a crucial role in the resistance responses to salt stress in both transgenic Arabidopsis and soybean plants, and it could be beneficial

in producing new transgenic plants with enhanced salt tolerance (*Li et al., 2021*). It has been determined that *MYB Domain Class Transcription Factors tMYB60* and *AtMYB96* play a role in the ABA signaling pathway, regulate response to drought stress, enhance resistance against diseases (*Seo & Park, 2010*), and are involved in stomatal movement (*Cominelli et al., 2005*) in Arabidopsis. *Su et al. (2010)* have also demonstrated the role of the MYB gene *OsMYBS3* in conferring cold tolerance in rice. MYB family genes, such as *R2R3-MYB* and *OsMYB2*, play a role in salt, cold, and dehydration tolerance (*Yang, Xiaoyan & Zhang, 2012*). Additionally, MYB-based transcriptome proteins were observed to actively participate in regulation to enhance tolerance to abiotic stress factors in the plant genome (*Roy, 2016*).

Plant *ABC transporter's* functions were first mentioned by *Linton (2007)* and play a critical role in transporting secondary metabolites, metals, hormones, xenobiotics, and pathogen responses in plants and in various aspects of plant development, all of which are important for global food security. However, plant *ABC transporters* are involved in the deposition and synthesis of secondary metabolites (*Theodoulou & Kerr, 2015*). Under greenhouse conditions, the overexpression of *Sucrose Synthase* (*SuSy*) genes in transgenic potatoes has increased starch content, UDP-glucose, and ADP glucose levels in tubers, enhancing yield (*Baroja-Fernández et al., 2009*). As a result, the increased activity of *SuSy* genes leads to starch accumulation in potato tubers, thereby increasing yield in potato plants (*Baroja-Fernández et al., 2009*). While the a-*sucrose synthase* gene *Sus4* significantly increased in the potato leaf and produced sucrose, the starch synthase gene *GBSS1* significantly increased in the sweet potato leaf and produced sucrose (slightly increased), fructose, and glucose (*Yoon et al., 2021*).

The study aimed to investigate the gene expressions of 10 potato genes (*MYB domain class transcription factor*, *NAC domain protein*, *ABC Transporter*, *StAREB2*, *StDHN1*, *StDREB1*, *StERD7*, *StHSP100*, *StRD22,* and *Sucrose Synthase* (*SuSy*) under drought stress in four cultivars and to explore the tolerance levels of potato cultivars under drought stress.

# MATERIALS AND METHODS

## Plant and cultivations

The commercial potato cultivars Brooke, Vr808, Orwell, and Sch909, which have different genetic characteristics and technological parameters from PepsiCo, and Frito Lay (https://fritolay.com.tr/) company were used. The experiment involved growing four plants in each 20-liter production pot. Humus soil, sand, and pearlite were utilized in a 2:1:1 ratio. Each pot was filled with sowing soil mix. The experiment was carried out on May 6, 2021, and finalized on August 4, 2021 (90 days). The plants were watered after sunset to reach maximum irrigation efficiency and increase water benefits. The fertilizer application to the pots has been calculated as the amount of fertilizer per plant with a calculation of 75,000 plants/ha. Before sowing, nitrogen (N)-Phosphorus and (P)-potassium (K) was added to potting soil mix. To prevent slime formation in the pots, reach the optimum infiltration rate, and provide complete drainage conditions, 1/4 of the fine sand was mixed into the potting soil. An insecticide commercially named Delegate 250 wg containing

25% Spinetoram as the active substance was used against potato tuberworm (*Phthorimaea operculella*), and a commercial fungicide called Phasma with 50 g/L Fludioxonil + 40 g/L Sedaxane was used against root-crown rot disease and wart. The pesticide used to combat the potato moth amounted to 24 g/da, and the pots were sprayed with 35 mL/100 g of this solution for seed diseases caused by *Rhizoctonia solani* on a wind-free afternoon.

## Method

This experiment was conducted according to completely randomized experimental design with four replications under greenhouse conditions. Once the plants reached the true leaf stage, thinning was initiated to leave one plant per pot. All the pots were well-watered until they reached field capacity (4,800 mL). Field capacity was calculated with the formula given below for the pots experiment;

Field capacity = (Weight of soil at maximum water holding-weight of oven-dried soil)/weight of oven-dried soil) (*Junker et al., 2015*).

As for field capacity, a total of 4,800 mL of water was given with the sowing (100% field capacity, Control). After 25 days from planting, when all the plants shoot emergence, drought stress treatments were initiated. Drought stress was performed as control = 100% field capacity, slight drought = 75% field capacity (3,600 mL), moderate drought = 50% field capacity (2,400 mL), and severe drought = 25% field capacity (1,200 mL).

## RNA extraction, quality testing, and complementary DNA (cDNA) synthesis

From each replication, a 100 mg leaf sample from young leaves which contain intense deoxyribonucleic acid (DNA) and reoxyribonucleic acid (RNA) were taken from each replication of drought stress, including control, were washed with distilled water (dH$_2$O) to remove the remains of any living creature, placed in dry ice at −80 °C, and transported by cold chain box to the laboratory to extract RNA (MST Laboratory; http://www.mstlab.com.tr/iletisim.php) From each leaf sample, the total RNA was isolated using an isolation kit (A.B.T., Blood/Tissue RNA Purification Kit for Leukemia). After RNA isolation, agarose gel electrophoresis was run to check the integrity of the RNAs in the Agilent 2100 Bioanalyzer. All RNA samples had a 28S:18S ratio within in range of 1.8−2.0 in intact of the 28S, 18S, and 5S RNAs bands, which were observed to be intact. In addition, RNA integrity Number (RIN) >8.0 was obtained, this value, with high purity is compulsory to construct a DNA library and sequence. The purity values (260/280) of the isolated RNAs were measured using a NanoDrop (Thermo Scientific, NanoDrop, 2000/2000c Spectrophotometers). The concentration of extracted RNAs changed between 66–730 ng/ μL. A cDNA kit (High-Capacity cDNA Reverse Transcription Kit) was used to synthase the cDNA. For each sample, a total volume of 10 μL was obtained by using 10X Reaction Buffer (2 μL), dNTP mix (2.5 mM each) (1 μL), random hexamer (50 μM) (2 μL), reverse transcriptase (200 U/ μL) (1 μL), RNase inhibitor (0.5 μL), RNase free water (3.5 μL), and the total volume was 20 μL. During the PCR step, the synthesis was carried out such that the temperature was 25 °C (5 min) during Step 1, 37 °C (120 min) during Step 2, 85 °C (10 min) during Step 3, and 4 °C (∞) during Step 4.

## Gene expression analysis using real-time PCR (RT-qPCR)

A real-time PCR (RT-qPCR) gene expression analysis consists of RNA isolation from the leaf sample, primer design and optimization, reverse transcription, cDNA synthesis, and amplification through RT-PCR (qPCR, quantitative PCR). The 10 genes obtained from previous studies listed in Table 1 were synthesized as primers. Then, RT-PCR was performed using a real-time PCR Master Mix (A.B.T, 2X qPCR SYBR-Green Master Mix with ROX). For each sample, the solution was filled up until the volume equaled 20 µL using Master Mix (10 µL, cDNA (10 ng µL) (2 µL), forward primer (10 µM) (1 µL), reverse primer (10 µM) (1 µL), and DNA-free purified water (6 µL) using RT-PCR (7500 Real-Time PCR System; Applied Biosystems, Waltham, MA, USA); subsequently, cDNA was added to the solutions. Complementary DNA was added last to the PCR plates to not affect the complementary DNA's structure. All other components were prepared as a mix and distributed to the PCR plate wells. RT-qPCR reaction processes were carried out according to the below protocol. Gene expression analyses were carried out in two phases as follows: the cycling stage at 95 °C (15 min), 95 °C (15 s), 60 °C (30 s), 72 °C (30 s ×40 cycles) and the melt curve stage at 95 °C (15 s); 60 °C (15 s), 95 °C (15 s) (calculated with an increase of +2% °C) and 60 °C (15 s). In this study, the St*Actin* gene was used as a housekeeping gene to calculate the gene expression (*Shi et al., 2016*; *Musse et al., 2021*; *Livak & Schmittgen, 2001*). The St*Actin* gene isolated from DM1-3,51 6R44 cultivar is *a Solanum tuberosum* actin-58 (LOC102582178) transcript variant X1, mRNA gene in NCBI (National Center for Biotechnology Information). At the end of qRT-PCR, the $2^{-\Delta\Delta Ct}$ method was used to calculate the relative gene expression level (*Livak & Schmittgen, 2001*). The genes were normalized to ''1.00'' for the control group, and comparisons were made over these values.

## Statistical analysis

Two-way ANOVA analysis was performed with the Brown-Forsythe test using GraphPad Prism version 8.0.0 software (*Motulsky, 2016*). To determine the kinship among the potato drought-responsive genes, the gene sequences provided from NCBI were used to perform cluster analysis with the Neighbor-Joinning method and using MEGA 11.0.13 (The Molecular Evolutionary Genetics Analysis; https://www.megasoftware.net/) (*Tamura, Stecher & Kumar, 2021*) version software. The graphs of gene expression of the *MYB domain C.T.F*, *NAC* domain *protein, ABC Transporter, StAREB2, StDHN1, StDREB1, StERD7, StHSP100, StRD22,* and *Sucrose Synthase (SuSy)* genes in all potato cultivars were prepared using a GraphPad Prism version 8.0.0. To indicate the significance of differences between drought stress means asterisks were used (*), accordingly, significant at * = ($p \le 0.05$) and highly significant at ** = ($p \le 0.01$), while not significant ($p > 0.05$) is indicated with ''ns''.

## RESULTS

The differences in gene expression means have been determined for genotypes (G), drought stresses (D), and drought × genotype (D × G) interaction. In Table 2, the mean differences in gene expressions of the *NAC domain protein, StAREB2, StDREB1, StDHN1,*

**Table 1  List of primers used for qRT-PCR.**

| Gene | Accession No /Gene ID | Sequence (5′–3′) | Amplicon Size (BP) | Reference |
|---|---|---|---|---|
| MYB Domain C. T. F | PGSC0003DMG400033043 | F-5′-TATCGGTCGATGAGGGTGGTA-3′<br>R- 3′-TCTGGCTTGAAATCAGGCAAA-5′ | 115 | Gong et al. (2015) |
| NAC Domain Protein | PGSC0003DMG400015342 | F- 5′-AAGCAACGGGAACGGATAA-3′<br>R- 3′-TGCGACAAAGCACCCATT-5′ | 192 | Gong et al. (2015) |
| StActin | XM_006345899 | F- 5′-GTGTGATGGTGGGTATGGGT-3′<br>R- 3′-GGCTTCAGTTAGGAGGACAGG-5′ | 200 | Musse et al. (2021) |
| StDREB2 | JN125858 | F- 5′-AAAGCAGAGGGAACACCAAC-3′<br>R- 3′-GGGAAGAATAAGAACCAAGCCA-5′ | 128 | Musse et al. (2021) |
| StDHN1 | XM_015304546 | F- 5′-AGGAGAAATTGCCAGGAGGT-3′<br>R- 3′-GTGCCTTCCATACCATAACCAG-5′ | 85 | Musse et al. (2021) |
| StAREB1 | XM_006346349 | F- 5′-GGCTCAAGGCGGAGTTATG-3′<br>R- 3′-GGGAAGGTGAAAGAGACGATG-5′ | 125 | Musse et al. (2021) |
| StERD7 | XM_006359626 | F- 5′-TGGGGATGTTACTGTGGATAGG-3′<br>R- 3′-GAGACCTTCACTACACCTGAGA-5′ | 180 | Musse et al. (2021) |
| StHSP100 | XM_006338326 | F- 5′-GCAAGTTTATGTTGACCAGCC-3′<br>R- 3′-GCCGTGTCTGAAATGCGA-5′ | 105 | Musse et al. (2021) |
| StRD22 | JX839749 | F- 5′-CACACAGTTAGCAAGAGCAAAG-3′<br>R- 3′-GGTATCCAAGTGACAAACAGCA-5′ | 93 | Musse et al. (2021) |
| Sucrose Synthase | BQ117791 | F- 5′- TTCAAGGATCGAAAGCCACG3′<br>R- 3′- ATGTATTCCCAGACACCAGGCC5′ | 190 | Buell et al. (2002) |
| ABC transporter | BQ514204 | F- 5′- C TCACAAGGTGGTGTTTCTGG -3′<br>R- 3′- CAACACCTCAGCTTCAAGTCG -5′ | 200 | Musse et al. (2021) |

*ABC Transporter, StERD7, MYB domain C.T.F, Sucrose Synthase, StHSP100*, and *StRD22* genes are highly significant ($p \leq 0.01$) for genotypes, drought stresses, and genotype × drought interaction.

The *Solanum tuberosum* L. drought-responsive genes have been divided into three main clusters based on ancestral (Fig. 1). The *Sucrose Synthase, StHSP100, StRD22*, and *StERD7* were clustered in the first main cluster, while the *StAREB1, MYB domain C.T.F, StDREB2, ABC transporter* genes clustered in the second main cluster. The third main cluster consists of *StDHN1* and *NAC domain* genes. The level of kinship between genes clustered in different branches increases as the distance increases. The weakest kinship was obtained between NAC domain protein and Sucrose synthase genes (Fig. 1).

The *SuSy* gene in the Brooke was expressed at all three drought stress (Fig. 2). The gene expression increased approximately 5.7-fold at slight, 1.1-fold at moderate, and approximately 4.3-fold at severe drought stages. In Vr808, under moderate drought stress, gene expression increased approximately 34-fold, but in slight and severe drought stress the gene showed a decreasing trend compared to the control group. The gene expression in the Orwell increased 18-fold at slight, 30-fold at moderate, and approximately 42-fold at the severe drought stress level. In the Brooke, the differences in gene expression between the control-slight, control-severe, slight-moderate, and moderate-severe were highly significant ($p \leq 0.01$), while the differences in the gene expression between control-moderate drought stress were found to be insignificant ($p > 0.05$). In Vr808, highly

Peer J

**Table 2  Two-way ANOVA of genes and drought stress in this study.**

| | | NAC domain Protein | StAREB2 | StDREB1 | StDHN1 | ABC Transporter | StERD7 | MYB domain C. T. F | Sucrose Synthase | StHSP100 | StRD22 |
|---|---|---|---|---|---|---|---|---|---|---|---|
| SOV | DF | | | | | Mean of Squares (MS) | | | | | |
| Drought (D) | 3 | 46.77** | 115.6** | 21.53** | 43.60** | 634.9** | 4,347** | 342.5** | 2,004** | 10.46** | 250,980** |
| Genotype (G) | 3 | 6.46** | 79.10** | 5.82** | 11.87** | 590.5** | 1,205** | 174.3** | 854.1** | 6.21** | 135,845** |
| D x G | 9 | 14.45** | 77.47** | 29.36** | 27.06** | 694.4** | 1,295** | 142.7** | 682.3** | 12.11** | 182,756** |
| Error | 64 | 0.23 | 0.28 | 0.28 | 0.28 | 0.28 | 0.28 | 0.28 | 0.28 | 0.28 | 0.28 |

**Notes.**

SOV, Source of variance; DF, Degrees of freedom.

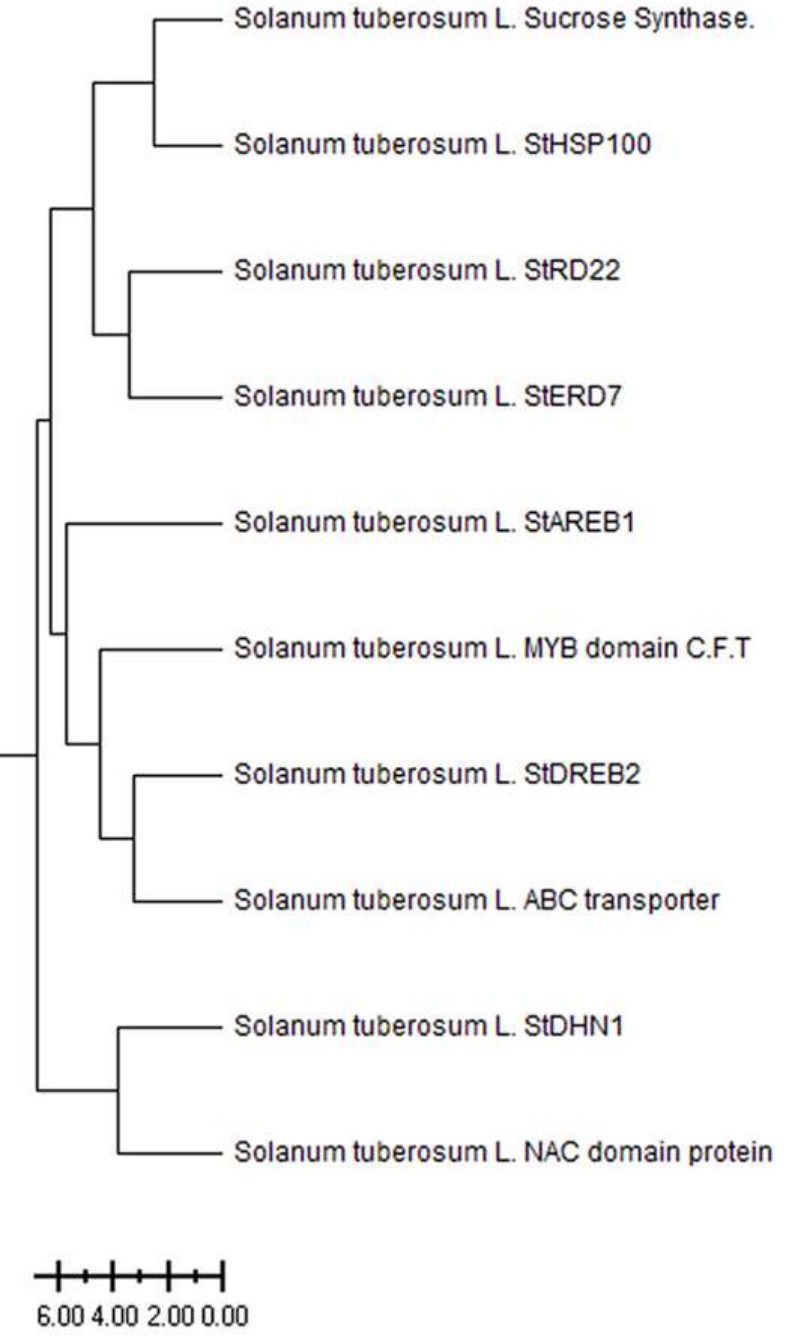

6.00 4.00 2.00 0.00

**Figure 1  Phylogenetic tree of patato drought-responsive genes.**

significant differences ($p \leq 0.01$) in *SuSy* gene expression level have been identified between control-moderate, slight-moderate, and moderate-severe, while insignificant differences ($p > 0.05$) of in expression levels of the gene have been identified between control-slight, control-severe, and slight-severe drought stress (Fig. 2). In Orwell, differences in gene expression between control-slight, control-moderate, control-severe, slight-moderate,

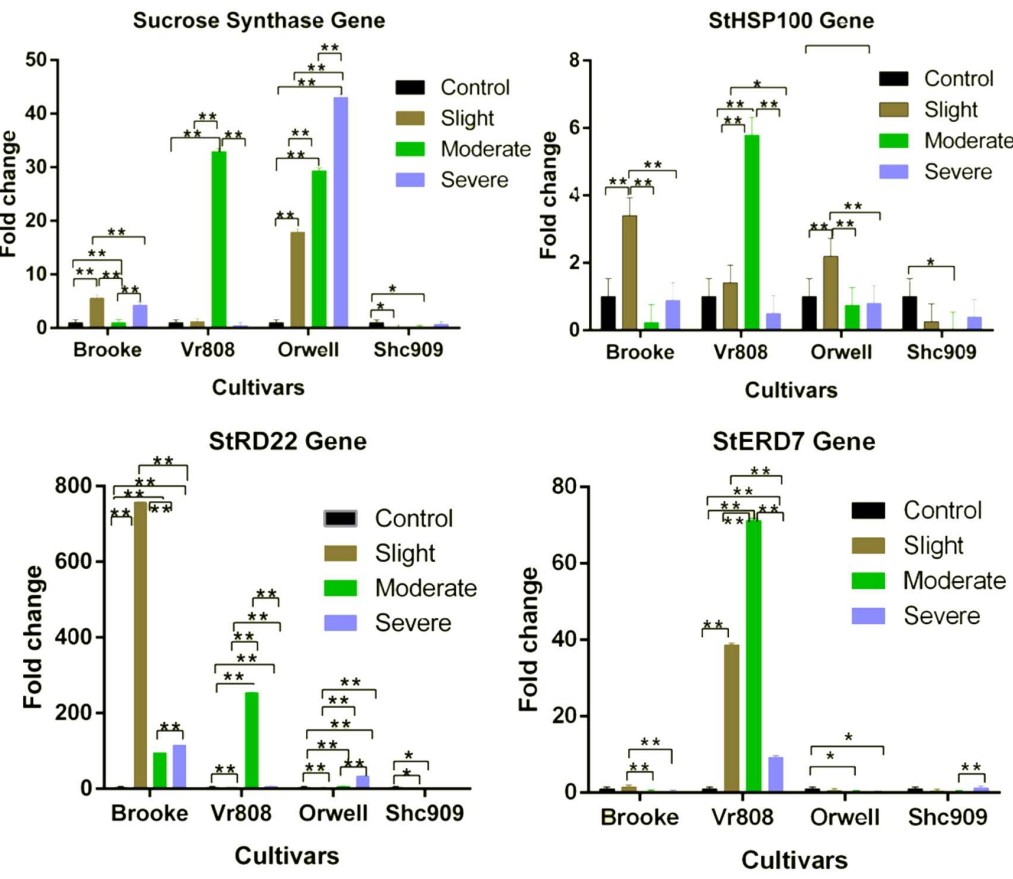

**Figure 2** Gene expression change in four potato genotypes under control and tree drought stress. The Relative gene expression changes of the *StERD7, StHSP100, Sucrose Synthase, and StRD22* genes in potato cultivars Brooke, Vr808, Orwell, and Shc909 are under Control (100 of Field capacity), Slight (75% of field capacity), Moderate (50% of field capacity), and Severe (25% of field capacity) drought stresses. Each colored graphic bar represents the fold change of gene expression at different irrigation regimes, while the $X$-axis represents the potato cultivars. 25 Days After Shoot Emergence (DASE) stress application was started. 65 days after stress application the leaf samples were picked up for RNA isolation. Two-way ANOVA analysis has been conducted at a confidence level of 95%. ** = Highly significant at $p \leq 0.01$, * = Significant at $p \leq 0.05$, and "ns" = Not significant ($p > 0.05$). Graphs and columns without asteriks show statistically insignificant differences in multiple comparisons.

slight-severe and moderate-severe statistically were highly significant ($p \leq 0.01$) (Fig. 2). Due to the lack of gene expression in the cultivar Shc909, the differences in gene expression between control-slight, and control-moderate drought stress were significant ($p \leq 0.05$), but in significant differences ($p > 0.05$) in the same gene expression level has been identified between control-severe, slight-moderate, slight-severe, and moderate-severe drought stress. The highest expression of the *SuSy* gene has been observed in the cultivar Orwell under severe drought stress, followed by the cultivar Vr808 under moderate drought stress. It has been observed that the cultivar Orwell is the most tolerant cultivar among potato cultivars (Fig. 2)

The *StHSP100* gene in the Brooke increased approximately 3.5-fold only at the slight drought stress but decreased under moderate and severe drought stress. A similar situation was observed for the cultivar VR808. Accordingly, the *StHSP100* gene in the cultivar Vr808 increased approximately 1.8-fold at slight, and 5.8-fold at moderate drought stress, but the gene decreased under severe drought stress. This is attributed to the high gene expression increase in Vr808 under moderate drought stress compared to the control. The gene, in the cultivar Orwell increased 2.3-fold at the slight, but decreased at moderate and severe drought stress. In Shc909, the same gene expression was decreased at slight, moderate, and severe drought stress (Fig. 2). The activities of the *StHSP100* gene in the cultivar Orwell were similar to the Brooke and Vr808. In Brooke, the differences in gene expression between control-slight, slight-moderate, and slight-severe were highly significant ($p \leq 0.01$), but the differences of the same gene expression level between control-moderate, control-severe, and moderate severe were insignificant ($p > 0.05$). In Vr808, highly significant differences ($p \leq 0.01$) in expression levels of the gene have been identified between control-moderate, slight-moderate, and moderate-severe. A significant difference ($p \leq 0.05$) in the gene expression level was between slight-severe drought stress, while insignificant differences ($p > 0.05$) in the gene expression was between control-slight and control-severe drought stress. In Orwell, the differences in gene expression between control-slight, slight-moderate, and slight-severe drought stress were highly significant ($p \leq 0.01$), however, insignificant differences ($p > 0.05$) in the gene expression has been identified between control-moderate, control-severe, and moderate-severe drought stress (Fig. 2). In the cultivar Shc909, the differences in gene expression between control-moderate were significant ($p \leq 0.05$), while insignificant differences ($p > 0.05$) in gene expression level have been found between control-slight, control-severe, slight-moderate, slight-severe and moderate-severe drought stress. In cultivar Vr808, this up-regulated more than other cultivars at moderate drought stress and made Vr808 the most tolerant cultivar and followed cultivar Brooke (Fig. 2).

In Brooke, approximately 780-fold overexpression was observed under slight, about 80-fold under moderate, and approximately 140-fold under severe drought stress. The gene has been observed as the gene with the highest expression increase among all genes under drought stress in Brooke. In Vr808, a notable gene expression increase of approximately 270-fold has been observed under moderate and about a 10-fold increase under severe drought stress. In the Fig. 2, the *StRD22* gene in Orwell has shown approximately a 35-fold increase at severe, about an 8-fold increase under moderate, and a 4-fold increase at slight drought stress. There was not any increase in this gene expression Shc909. In Brooke, differences in the expression level of the *StRD22* gene between control-slight, control-moderate, control-severe, slight-moderate, slight-severe, and moderate-severe drought stress were highly significant ($p \leq 0.01$). In cultivar Vr808, highly significant differences ($p \leq 0.01$) in expression levels of the gene have been identified between control-slight, control-moderate, control-severe, slight-moderate, slight-severe, and moderate-severe drought stress. In Orwell, differences in expression levels of the gene between control-slight, control-moderate, control-severe, slight-moderate, slight-severe, and moderate-severe drought stress were highly significant ($p \leq 0.01$). In Shc909, significant differences

($p \leq 0.05$) in the gene expression have been identified between control-slight, and control-moderate drought stress, but insignificant differences ($p > 0.05$) in gene expression level have been identified between control-severe, slight-moderate, slight-severe, and moderate-severe drought stress (Fig. 2). Accordingly, this gene has been excessively expressed in the cultivar Brooke at slight, moderate, and severe stresses compared to the control, highlighting that cultivar Brooke is the most tolerant. Following this, the cultivar Vr808 has shown overexpression under moderate drought stress, indicating its tolerance as well.

The *StERD7* gene increased 1.5-fold in the Brooke at the slight drought stress but decreasing gene expression was observed at moderate and severe drought stress. In the cultivar Vr808, the gene has shown approximately a 40-fold increase in slight, 70-fold increase in moderate, and about a 10-fold increase at severe drought stress. While the same gene expression decreased in the cultivar Orwell at the three drought stresses. Similar gene activity was obtained in the cultivar Shc909 at slight and moderate drought stress but at severe drought stress about a 1.2-fold increase was observed compared to the control group (Fig. 2). In Brooke, the differences in gene expression between slight-moderate, slight-severe were significant ($p \leq 0.01$), while the difference in gene expression between control-slight, control-moderate, control-severe, and moderate-severe was in significant ($p > 0.05$). In the cultivar Vr808, the expression difference between control-slight, control-moderate, control-severe, slight-moderate, slight-severe, and moderate-severe was highly significant ($p \leq 0.01$). In Orwell, the difference in gene expression between control-moderate and control-severe was significant ($p \leq 0.05$), but while insignificant differences ($p > 0.05$) in gene expression level have been identified between control-slight, slight-moderate, slight-severe, and moderate-severe drought stress. In Shc909, the differences in gene expression between control-slight, control-moderate, control-severe, slight-moderate, and slight-severe were in significant ($p > 0.05$), while the highly significant ($p \leq 0.01$) differences in the gene expression level have been identified between moderate-severe drought stress (Fig. 2). In Fig. 2, the *StERD7* gene overexpressed in cultivar Vr808 and made this cultivar the most tolerant. It has been observed that cultivar Vr808 exhibited the highest tolerance under all three drought stress conditions.

While the *StDREB1* gene in the cultivars Brooke, VR808, and Orwell increased under different water stress. However, a decreasing gene expression was observed in the Shc909 at tree drought stress (Fig. 3). The gene in the Brooke increased approximately 7-fold at the slight but decreased in moderate and severe. In Vr808, increased approximately 4-fold at the moderate drought level but decreased gene expression in slight and severe drought stress. The expression of the gene has gradually increased in the cultivar Orwell, showing an approximately 1-fold increase in slight, 2-fold increase in moderate, and about 6.6-fold increase in severe drought stress (Fig. 3). In Brooke, the differences in gene expression at control–slight, slight–moderate, and slight-severe were highly significant ($p \leq 0.01$), while insignificant ($p > 0.05$) differences in gene expression between control-moderate, conrol-severe, and moderate-severe were observed. In the cultivar Vr808, the differences in gene expression between control–moderate, slight–moderate, and moderate–severe drought stress were highly significant ($p \leq 0.01$), but insignificant ($p > 0.05$) differences in gene expression between control-slight, control-severe and

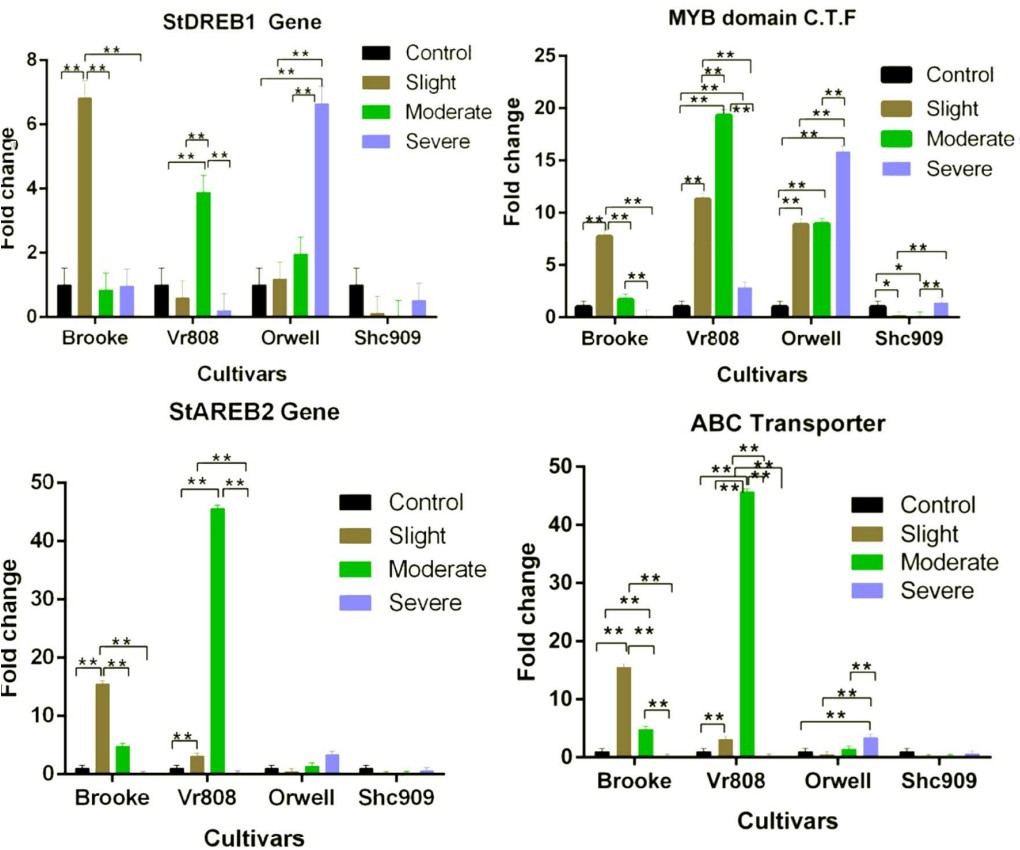

**Figure 3  Gene expression change in four potato genotypes under control and tree drought stress.** The expression changes of the *MYB domain C.F.T, ABC Transporter, StAREB2*, and *StAREB1* genes in potato cultivars Brooke, Vr808, Orwell, and Shc909 are under control (100 of field capacity), slight (75% of field capacity), moderate (50% of field capacity), and Severe (25% of field capacity) drought stresses. Each graphic bar represents the fold change of gene expression at different irrigation regimes, while the *X*-axis represents the potato cultivars. Twenty-five days after shoot emergence (DASE) stress application was started. Sixty-five days after stress application the leaf samples were picked up for RNA isolation. Two-way ANOVA analysis has been conducted at a confidence level of 95%. ** = Highly significant at $p \leq 0.01$, * = Significant at $p \leq 0.05$, and "ns" = Not significant ($p > 0.05$). Graphs and columns without asteriks show statistically insignificant differences in multiple comparisons.

slight-severe were obtained (Fig. 3). Also, in the cultivar Orwell, the differences in gene expression between control-slight, control-moderate, and slight-moderate was insignificant ($p > 0.05$), and the differences between control-severe, slight-severe and moderate-severe were highly significant ($p \leq 0.01$) (Fig. 3). In the cultivar Shc909, the differences in gene expression were insignificant ($p > 0.05$). In Fig. 3, the cultivar Orwell has been identified as the most tolerant under severe drought stress in the gene followed by cultivar Vr808 under moderate drought stress.

In the cultivar Brooke, the *MYB domain class transcription factor* exhibited an approximately eight-fold increase at slight drought stress and a two-fold increase at moderate, but a decrease was observed at severe (Fig. 3). The gene, in the cultivar Vr808

has shown an approximately 11-fold increase at slight, approximately 18-fold increase at moderate, and about a four-fold increase at severe drought stress. In the cultivar Orwell, the gene expression increased approximately nine-fold at slight, and moderate but, the same gene increased about 16.5-fold in severe drought stress (Fig. 3). In cultivar Shc909, the gene expression at slight and moderate decreased but increasing gene expression about 3.5-fold was obtained at severe drought level. In the Brooke, the differences in gene expression between control-slight, slight-moderate, and slight-severe and moderate-severe drought were highly significant ($p \leq 0.01$), while insignificant differences ($p > 0.05$) in gene expression level has been identified between control-moderate, control-severe (Fig. 3). In the cultivar Vr808, the differences in gene expression between control-slight, control-moderate, control-severe, slight moderate, slight-severe, and moderate-severe were highly significant ($p \leq 0.01$). In Orwell, differences in gene expression level of the gene between control-slight, control-moderate, control-severe, slight-severe, and moderate-severe were highly significant ($p \leq 0.01$), while the differences in gene expression between slight-moderate drought stress were insignificant ($p > 0.05$). In the Shc909, the gene expression level between control-slight, and control-moderate was significant ($p \leq 0.05$), the differences in gene expression between slight-severe, and moderate-severe were highly significant ($p \leq 0.01$), while insignificant differences ($p > 0.05$) in the same gene expression was found between control-severe and slight-moderate drought stress (Fig. 3). The highest gene expression was obtained in cultivar Vr808 at moderate, but because of the cultivar Orwell exhibits a close gene expression value at severe drought stress, Orwell was accepted as the most drought-tolerance for the *MYB domain C.T.F.* (Fig. 3).

The *StAREB2* gene expression in the cultivar Brooke increased 15-fold at the slight, 5-fold at moderate, but decreased in severe drought stress compared to the control. The same gene was expressed approximately 6.5-fold at the slight and 47-fold at the moderate drought stress in Vr808, but no gene expression at the severe drought level (Fig. 3). In Orwell, the gene started to increase at moderate stress about 1.2-fold and 1.4-fold at severe drought stress. Decreasing gene expression was observed at slight and severe drought stress levels, while no gene expression was observed in moderate drought levels, in cultivar Shc909 (Fig. 3). In Brooke, the differences in gene expression between control-slight, slight-moderate, and slight-severe were highly significant ($p \leq 0.01$), and between control-moderate, control-severe, and moderate-severe drought stress was insignificant ($p > 0.05$). In Vr808, the differences in gene expression between control-slight and control-moderate, slight-severe and moderate-severe were highly significant ($p \leq 0.01$), but between control-severe and slight-severe drought stress were insignificant ($p > 0.05$). In cultivars Orwell and Shc909, differences in gene expression between drought stress and control-drought stress were insignificant ($p > 0.05$). The highest gene increase in gene expression has been observed in the cultivar Vr808 at moderate drought stress and made the cultivar Vr808 the most tolerant one. Under slight drought stress, the cultivar Brooke has also exhibited a high tolerance to drought (Fig. 3).

The *ABC Transporter* gene in the cultivar Brooke increased 15-fold at the slight drought and approximately 4.8-fold at the moderate drought stress but decreased at the severe drought level. The gene in the Vr808 was expressed approximately two-fold at the slight

and 45-fold at the moderate drought stress, but no gene expression at severe drought stress. The gene in the cultivar Orwell decreased at the slight drought level compared to the control and increased approximately 1.5-fold at the moderate and 3.5-fold at the severe drought stress (Fig. 3). In cultivar Shc909 gene expression was decreased compared to the control. In cultivar Brooke, the differences in gene expressions between control-slight, control-moderate, slight-moderate, slight-severe, and moderate-severe drought stresses were highly significant ($p \leq 0.01$), but insignificant differences ($p > 0.05$) in gene expression level have been identified between control-severe drought stress. In Vr808, the differences in gene expression between control-slight, control-moderate, slight-moderate, slight-severe, and moderate-severe were highly significant ($p \leq 0.01$), but insignificant differences ($p > 0.05$) in gene expression levels have been identified between control-severe was insignificant ($p > 0.05$). In Orwell, the differences in gene expression between control-severe, slight-severe, and moderate-severe were highly significant ($p \leq 0.01$), while insignificant differences ($p > 0.05$) in gene expression levels have been found between control-slight, control-moderate, and slight-moderate drought stress (Fig. 3). Statistically, there were no significant differences ($p > 0.05$) in gene expression in the cultivar Shc909. The cultivar Brooke has shown an increase in tolerance under slight drought stress, but at the severe drought level, its tolerance has been lost. This gene has significantly increased the tolerance of the cultivar Vr808 under moderate drought stress. This overexpression of the gene in cultivar Vr808, makes this cultivar as the most tolerant cultivar. The cultivar Orwell exhibited tolerance under severe drought stress compared to the control (Fig. 3).

Gene expression analysis showed that the *StDHN1* gene was expressed in four cultivars. The gene in Brooke started to increase at slight, decreased to express in moderate, and 4-fold increased at severe drought stress; while increased in Vr808 approximately 2.2-fold at slight, 4-fold at moderate, and expressed approximately 3.6-fold at severe (Fig. 4). In Orwell, the gene expression was increased approximately 8.4-fold at slight but decreased at moderate and severe drought stress. In Shc909, the gene increased approximately 5-fold at slight but decreased in remaining drought stress levels (Fig. 4). In the cultivar Brooke, the differences in gene expression at control-severe, slight-severe, and moderate-severe drought stress were highly significant ($p \leq 0.01$), but an insignificant difference in gene expression was found between control-slight, control-moderate, and slight moderate drought stress ($p > 0.05$) (Fig. 4). In Vr808, the differences in gene expression between control-moderate, control-severe, and slight-moderate were highly significant ($p \leq 0.01$) (Fig. 4), while the insignificant difference in gene expression was between moderate-severe drought stress ($p > 0.05$) In Orwell, while the differences in gene expression between control-slight, slight–moderate, and slight–severe was highly significant ($p \leq 0.01$), insignificant differences ($p > 0.05$) in gene expression between control-moderate, control-severe and moderate-severe drought stress was observed (Fig. 2). in cultivar Shc909, the differences in gene expression between control-slight, slight-moderate, slight-severe was highly significant ($p \leq 0.01$), but insignificant ($p > 0.05$) differences in gene expression was between control-moderate, control-severe and moderate-severe (Fig. 4). In Fig. 4, the highest up-regulation under slight drought stress has occurred in the cultivar Orwell. The level of gene expression

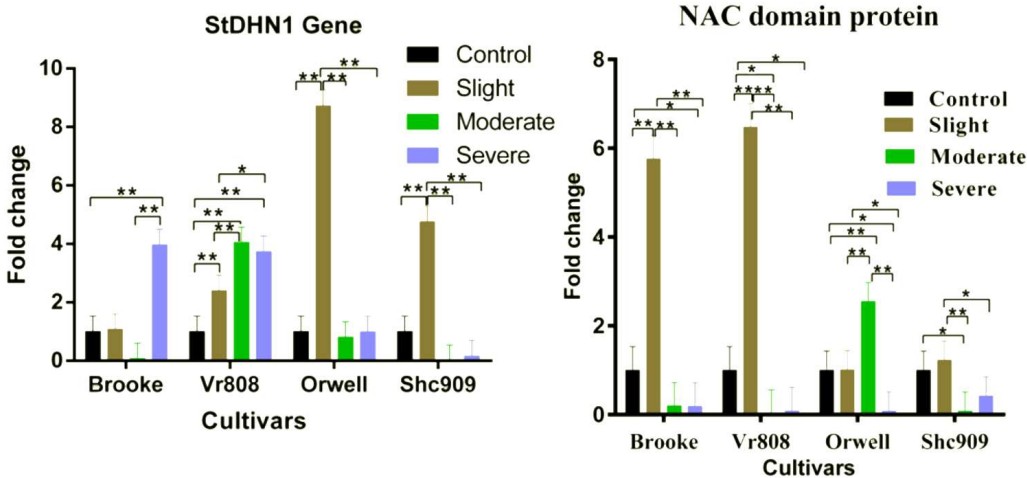

**Figure 4 Gene expression change in four potato genotypes under control and tree drought stress.** The expression changes of the *NAC domain protein* and *StDHN1* genes in potato cultivars Brooke, Vr808, Orwell, and Shc909 are under control (100 of field capacity), slight (75% of field capacity), moderate (50% of field capacity), and severe (25% of field capacity) drought stresses. Each graphic bar represents the relative gene expression fold change at different irrigation regimes, while the *X*-axis represents the potato cultivars. Twenty-five days after shoot emergence (DASE) stress application was started. Sixty-five days after stress application the leaf samples were picked up for RNA isolation. Two-way ANOVA analysis has been conducted at a confidence level of 95%. ** = Highly significant at $p \leq 0.01$, * = Significant at $p \leq 0.05$, and "ns" = Not significant ($p > 0.05$). Graphs and columns without asteriks show statistically insignificant differences in multiple comparisons.

increases in severe drought stress in the cultivar Brooke, and Vr808 has revealed that these cultivars are more tolerant to drought.

In Fig. 4, the *NAC domain protein* gene increased approximately 6-fold at the slight drought stress in the Brooke compared to the control group and increased approximately 6.5-fold at the slight drought stress in the Vr808. The gene expression was decreased under other stress conditions in Brooke and Vr808. In Orwell, the gene has shown approximately a 2.6-fold increase only under moderate drought stress and decreased in remaining drought stress. In the cultivar Shc909, except for slight drought stress, gene expression was decreasing at moderate and severe drought levels. In the cultivar Brooke, the differences in gene expressions between control-slight, slight-moderate, and slight-severe drought stress was highly significant ($p \leq 0.01$), while insignificant differences were between control-moderate and moderate-severe. Similar observations have been made in cultivar Vr808. In Vr808, differences in gene expression between control-slight, slight-moderate, and slight-severe drought were highly significant ($p \leq 0.01$), between control-moderate and control-severe was significant ($p \leq 0.05$), but no significant difference in the gene expression between moderate-severe drought stress was obtained ($p > 0.05$). In Orwell, the differences in gene expression between control-moderate, slight-moderate and moderate-severe drought stress were highly significant ($p \leq 0.01$), while a significant difference ($p \leq 0.05$) was obtained between control-severe and slight-severe drought stress. Statistically Insignificant gen expression between control-slight was observed ($p > 0.05$). In the cultivar Shc909,

the difference in gene expressions between slight-moderate drought stress was highly significant ($p \leq 0.01$), while the gene expression difference between control-moderate and slight-severe drought stress was observed significant ($p \leq 0.05$). differences in gene expression between control-sligh, control-severe and moderate-severe drought stress were insignificant ($p > 0.05$) (Fig. 4). In Fig. 4, the gene has significantly up-regulated, enhancing the drought tolerance of the cultivar Vr808 at slight drought stress, followed by the cultivar Brooke under the same drought stress. The same gene has been observed to increase the drought tolerance of the cultivar Orwell under moderate drought stress.

## DISCUSSION

The measurement reactions of potato plants under water stress, and discovering genetic mechanisms under drought conditions, play an effective role in combating the global problem of drought. In this study, 10 drought-responsive genes were used in the potato plant, and the reactions of four potato cultivars to drought were examined. Additionally, the gene expressions of these drought-responsive genes under different drought stress conditions were investigated. Studies show that Sucrose enhances resistance in plants against various biotic and abiotic stress factors, especially drought (*Lv et al., 2008*). Additionally, the sucrose content increased the tolerance of wheat plant under drought stress (*Kutlu et al., 2021*). Transcript level of *SuSy* significantly increased almost all drought stress in four potato cultivars. Additionally, the difference in gene expression between drought stress was found to be significant almost in all cultivars. Similar statistical significance was reported in Sugarcane under water deficit (*Iskandar et al., 2011*), in potato under water stress (*Kondrák et al., 2012*), and in the Sugarcane plant (*Papini-Terzi et al., 2009*). Parallel to our study, *Wang et al. (2022)* have demonstrated that moderate and severe drought stress decreases SuSy gene expression and reduces the sucrose content in the leaves of the plant under prolonged drought. *Yu et al. (2016)* also determined the *Sucrose Synthase* gene activity under drought stress. *Wang et al. (2022)* observed that the activity of the *Sucrose Synthase* gene decreased as drought increased, with a reduction of up to 53.84% compared to the control under severe drought stress. In light of the findings in this study and the findings in the literature, the increase in the transcription of the *SuSy* gene under drought stress and the increases in mRNA levels and statistically significant differences between different drought levels, it was observed that this gene increases drought tolerance, especially in potato plants. Additionally, water deprivation heightened the activities of sucrose metabolic enzymes and increased the expression of genes such as SuSy, INV, and SPS. Furthermore, the expression levels of SWEET and SUC were up-regulated under drought stress, facilitating the transport of sucrose from source to sink (*Thomas & Beena, 2021*).

Heat shock proteins (HSPs) play a role in the intracellular transport, remodeling, assembly, and denaturation of proteins to increase plant tolerance under abiotic stress conditions (*Hasanuzzaman et al., 2013*). *Panzade et al. (2021)* conducted a gene expression analysis to identify the regulation of the *HSPs* gene family members such as *ZJHsp70* and *jHSP100* to drought, heat, and salinity stress and the analysis resulted in not all *Hsp70*

genes (*Noël et al., 2007*). This indicates that the tolerance of these cultivars to drought decreases proportionally with the severity of drought. In the cultivar Shc909, there was a decrease in gene expression in response to drought, suggesting a considerably low tolerance of this cultivar to drought (Fig. 2). In this study, almost, the differences in gene expression between drought stress were significant in cultivar, but insignificant differences in gene expression between control-slight, and control-moderate in cultivar Brooke were obtained. With increasing the drought, a highly significant differences in gene expression reporterd by *Juneja et al. (2023)*. *Singh et al. (2016)*, *Chaudhary et al. (2019)* and *Liu et al. (2018)* stated that *Hsp100* genes is a key for increasing the tolerance of plants under drought stress, such as drought, heat, and salinity. *Juneja et al. (2023)* reporterd an enormously mRNA up-regulation in non-primed but a down-regulation of primed plants. Unlike our results, the gene expressions of primed plants are low under drought stress, while the gene expressions of non-primed plants are high. It is predicted that plants subjected to priming early growth stage are protected by priming from prolonged drought.

The *StRD22* gene encodes a regulatory protein (responsive to desiccation 22) is from the protein-coding *RD22* gene family, increasing plant tolerance under salinity, drought, and heat stresses (*Byun, Kwon & Park, 2007*). The *StRD22* gene is one of the heat shock proteins involved in the response of plants to regulate adaptation to stress factors (*Musse et al., 2021*). In the cultivar Shc909, the gene down-regulated. Accordingly, at the beginning of the drought, Brooke exhibited a high tolerance to drought, but as the severity of the drought increased, its tolerance decreased. In this study, almost, the differences in gene expression between drought stresess were significant in cultivar, but insignificant differences in gene expression between control-slight, and control-moderate in cultivar Shc909 was obtained. *Ma et al. (2021)* reported similar results that the expression of ABA-responsive genes including the *sTrd22* gene under Polyethylene glycol (PEG) drought stress up-regulated significantly. Orwell, on the other hand, demonstrated tolerance to drought from the onset of drought stress, reaching maximum tolerance under severe drought conditions. The cultivar Vr808 showed high tolerance to drought in moderate drought stress, but decreased gene activity was observed in the *sTrd22* gene to increase tolerance to drought at other stress levels (Fig. 2). At the same time, significant changes have been observed in the gene transcription levels at the severe drought stress level compared to the control (*Musse et al., 2021*).

The EARLY RESPONSIVE TO DEHYDRATION 7 (ERD7) gene is up-regulated under biotic and abiotic stress and commonly used as a marker in plant stress response (*Rasmussen et al., 2013*; *Cheng et al., 2013*). The gene expressions of chaperone proteins (Heat shock protein) such as *StDHN1*, *StTAS14*, *StERD7*, *StRD22*, and *StHSP100*, under drought stress conditions using RT-PCR, and reported significant increases in transcription under severe drought stress compared to mild drought stress and control (*Musse et al., 2021*). The transported genes may increase the drought tolerance ability in plants. Among all potato cultivars the most drought tolerance cultivar was Vr808 (Fig. 2). The increase in the Vr808 cultivar in slight and moderate under drought stress and the decrease in severe suggest that the plant responds positively to drought stress by increasing transcription (Fig. 2). However, under severe drought stress, it can be said that the plant attempts to

resist drought but may ultimately lose the battle. Highly significant differences in gene expressions were obtained between drought stress almost in all cultivars, in a study, the genes, including *StERD7*, *StDHN1*, *StTAS14*, *StRD22*, and *StHSP100*, exhibited statistically significant induction in transcription levels in SDW compared to control and MDW (*Musse et al., 2021*). Additionally, *Janiak, Kwaśniewski & Szarejko (2016)* stated signficant differences in gene expression in roots under drought stress. (*Molina et al., 2008*; Molina et al., 2011; *Venu et al., 2013*) reported such changes in gene expression in leaves and roots under water deficit or salt stress. Similar gene expression distributions between drought stress were reported.

The *StDREB1* gene, which was overexpressed in this study, also responded similarly under drought stress in tobacco plants (*He et al., 2022*). It was revealed that the *StDREB1* gene was overexpressed in the tobacco cultivar and that the gene was transferred and increased growth and development under drought stress in transgenic tobacco genotypes (*He et al., 2022*). Similarly, *Bouaziz et al. (2013)* revealed that overexpression of the *StDREB1* gene increases tolerance to salt and drought stress in transgenic potato plants and activates stress-responsive genes and others, such as calcium-dependent protein kinases. The authors claimed that the increase of this tolerance is probably related to the increase in P5CS-RNA (∆ 1-Pyrroline-5- Carboxylate Synthetase) gene expression, and it may be associated with the accumulation of the proline osmoprotectant used to reduce water loss. In wheat, the *Arabidopsis AtWRKY30* transcription factor increased the tolerance (*El-Esawi et al., 2019*). The observation of the *StDREB1* gene expression level is high in some cultivars under drought stress while decreasing gene expression is observed in others under drought stress, suggesting the possibility of an environmental effect on gene expression without altering the gene sequence. This situation evokes the concept of epigenetics.

Transcription factors (TFs) such as *MYB* (*Shin et al., 2011*), *WRKY* (*Ren et al., 2010*), *ERF* (*Zhang et al., 2009*), and *bHLH* play a critical role in signal transduction pathways involved in the response of plants to drought stress (*Shinozaki & Yamaguchi-Shinozaki, 2007*; *Golldack, Lüking & Yang, 2011*). Additionally, *Roy (2016)* reported that TFs play a key role in gene expression. On the contrary of cultivars Vr808 and Orwell, a downregulation under drought stress was reported in fiber elongation gene expression (*Padmalatha et al., 2012*). Downregulation was reported by *Pieczynski et al. (2013)* in potatoes and *Zhao et al. (2013a)* in tomatoes. *Yu et al. (2019)* reported a statistically significant difference in gene expression between drought stress and diverse gene expressions under drought stress in different tissue of wheat cultivars, but under prolonged drought, only the transgenic cultivars survived.

In contrast to our study, it can be said that the ability of transgenic plants to survive under prolonged drought stress and the stable increase in gene expression is attributed to the transfer of genes responsible for drought resistance to these plants.

The *StAREB2* gene with the PGSC0003DMG400008011 gene ID is located on chr.4 (*Liu et al., 2019*). There are four *StAREB2* derivative genes, *StAREB1*, *StAREB2*, *StABI5*, and *StAREB4*, classified in the A-subgenomic group of potatoes, were distributed on the Chr.1, Chr.14, Chr.19, and Chr.11, respectively. The genes identified in *StAREB4*, *StAREB1*, and *StAREB2* each have five exons, which are the encoding regions of the gene, and four exons

of *StABI5* and *StAREB3* (*Liu et al., 2018*). Similar results obtained in a previous study was conducted by *García et al. (2012)*. A highly significant differences ($p \leq 0.01$) in gene expression between drought stress in cultivars VR808 and Brooke. Parallel to our results, *Musse et al. (2021)* reported that the *StAREB2* gene showed upregulation at severe Water Deficit (SWD) compared to control and Mild Water deficit (MDW), but no significant regulation was observed at MDW. In Fig. 3, the gene did not show any significant induction at any drought stress in Orwell and Shc909. *Musse et al. (2021)* stated that the *StAREB2* gene didn't show any significant gene expression differences in any drought stress. *Wang, Wang & Zhang (2013)* observed low gene expressions in the root system and leaves of the cotton plant. The difference between their findings and ours may be attributed to the fact that gene expression analyses were conducted during a period of prolonged drought stress compared to our study. The tolerance levels of the cotton varieties used against drought stress may also contribute to variations in gene expressions. Analyses conducted at different times when drought stress is more severe could have decisive effects on the differences in gene expressions as well. In the Arabidopsis plant, highly stable of *PP2A* and *EF1A* gene expressions were confirmed by *Czechowski et al. (2005)* under abiotic stress conditions.

*ABC transporters* perform indispensable functions in facilitating the transmembrane transport of diverse molecules, enabling adaptation to swiftly changing environmental conditions, including water deficit, heavy metal stress, and pathogen stress (*Martinoia et al., 2002*). In the *Arabidopsis* plant, *ABC transporters* play a key role in stomatal regulation under drought stress (*Kuromori, Sugimoto & Shinozaki, 2011*). Similar significance in gene expression was  reported by *Szalonek et al. (2015)* and *Pieczynski et al. (2018)*. *Chen et al. (2018)* stated that overexpression of the *TsABCG11* gene, a member of the *ABC transporter* gene increases the abiotic stress in the *Arabidopsis* plant. In a study of *Gossypium hirsutum* L. cotton species, *Selvam et al. (2009)* found that the *ABC transporter* gene plays a part in increasing stress tolerance in plants and animal. It was revealed that, unlike normal *ABC transporter* genes, the mutant gene *AtABCG22* in Arabidopsis increases water loss through transpiration and increases drought sensitivity in the plant (*Kuromori, Sugimoto & Shinozaki, 2011*). Nevertheless, it has been discovered that certain mutated *ABC transporter* genes function oppositely (*Kuromori, Sugimoto & Shinozaki, 2011*). The mutated *ABC transporter* gene may perform the opposite function of the normal *ABC transporter* gene due to mutations arising from recombinant DNA. In addition, overexpression of the *Arabidopsis GRF9* gene increases the plant drought tolerance but deletion of this gene causes a weak root system and inhibits growth (*He et al., 2015*).

The *StDHN1* gene uses 474 nucleotides and 157 amino acids to code protein and is an ORF (gene open reading frame) gene (*Charfeddine et al., 2017*). The *StDHN1* gene was expressed in all potato tissues at varying levels, the gene expression was higher in stems and roots compared to leaves, but it was concluded that the *StDHN1* mRNA is abundant in roots (*Charfeddine et al., 2015*). Brooke and Vr808, although it shows different levels of increase under different drought stresses for four cultivars. (Fig. 4). *Chen et al. (2019)* reported a 5-fold increase in StDHN1 gene expression, demonstrating the highest level of induction, aligning with the findings of our study. In our study, the differences in the expression of this gene are highly significant ($p \leq 0.01$) among almost all drought stresses.

In contrast, *El-Esawi et al. (2019)* found the differences in gene expressions among drought stresses to be less significant ($p \leq 0.05$) in their studies. This difference is likely attributed to variations in the tolerance of plants to different drought stresses. The same less significant differences ($p \leq 0.05$) of potato Anexxin *STANN1* gene reported by *Szalonek et al. (2015)* in potato plant under drought stress. These differences may come from the environmental effect, period of time under drought stress, or the *STANN1* gene regulation speed under drought stress. In cultivar Shc909, similar to *Schafleitner et al. (2007)*, insignificant gene expression was between control-moderate ($p > 0.05$).

A protein-coding gene type, the *NAC (No Apical Meristem) domain protein* gene which is located on Chr.12 with PGSC0003DMG400015342 gene ID caused significant gene expression differences in four cultivars constitutes one of the largest gene families and is one of the 117 sequenced *NAC Domain protein* genes in *Arabidopsis thaliana* (*Le et al., 2011*; *Rushton et al., 2008*). At the beginning of drought stress, the *NAC domain protein* gene increased the Brooke and Vr808 drought tolerance but under prolonged drought stress, this cultivar drought tolerance capability was decreased. Previous studies indicated that these gene expressions increase the drought response in plants (*Wu et al., 2009*; *Nakashima et al., 2007*; *Chen et al., 2014*). In Orwell, there has been an increase in gene expression of the *NAC domain gene* associated with the increase in drought; however, a decrease in the quantity of this gene has been observed during the period of the most severe drought (Fig. 4). The fact that the cultivar Orwell is tolerant to drought can be associated with its high proline content. Similar studies approved that up-regulation of a gene enhances the tolerance under drought by causing a higher proline content (*Bandurska et al., 2017*; *Wang et al., 2018*). A higher gene expression of the *P5CS*, and drought stress tolerance were obtained under drought stress in wheat (*Dudziak et al., 2019*) and in Barley (Brandurska et al. 2019). Similarly, *Lu et al. (2012)* reported that a member from the NAC gene family *ZmSNAC1* increased the tolerance level of the plant under drought and low-temperature stress. Some genes such as *ZmSNAC1* (*Lu et al., 2012*), *TaNAC2a*, *TaNAC6a*, and *TaNAC4a* (*Tang et al., 2012*; *Xia et al., 2010*), *TaNAC69* (*Xue et al., 2011*), *RhNAC2* or *RhEXPA4* (*Dai et al., 2012*), *miR319*, *AsNAC60* (*Zhou et al., 2013*), *CsNAM* (*Paul et al., 2012*) and *SiNAC* (*Puranik et al., 2011*) were increased under salinity and drought stress. NAC TFs gene regulates many genes by binding the CATGTG sequence motif to activate transcription in response to drought (*Nakashima et al., 2007*). *Chen et al. (2011)* reported a stress-responsive gene called *SNAC1*.

## CONCLUSION

It was observed that the cultivar Orwell was the most drought-tolerant among four potato cultivars since the *StRD22* was the highest up-regulated gene. Among all the genes most associated with drought was the *StRD22* gene, belonging to the *RD22* gene family; it was found to have increased tolerance to abiotic stress factors. The *MYB domain*, *StERD7*, *SuSy*, *ABC transporter*, and *StDHN1* genes followed the *StRD22* gene in terms of tolerance levels to drought stress, respectively. In conclusion, it is strongly recommended that cultivars Orwell, Vr808, and Brooke can be used as parents in the development of drought-resistant

cultivars through rapid breeding programs such as Marker-assisted selection (MAS) and high throughput sequencing platforms such as genotyping-by-sequencing (GBS) and next-generation sequencing (NGS). In addition, using *StRD22*, *MYB domain*, SuSy, *ABC transporter*, and *STDHN1* genes to measure the response of other potato cultivars under drought stress is also encouraged.

## ACKNOWLEDGEMENTS

I am thankful to PepsiCo Frito Lay company for providing me with potato seed materials for the trials, and I would like to thank the agriculture engineer Hasan EZEL for supporting me during the trials and while obtaining the data.

### Funding
The author received no funding for this work.

### Competing Interests
The author declares that there are no competing interests.

### Author Contributions
- Sadettin Çelik conceived and designed the experiments, performed the experiments, analyzed the data, prepared figures and/or tables, authored or reviewed drafts of the article, and approved the final draft.

### Data Availability
    The raw data are available in the Supplementary File.

### Supplemental Information
Supplemental information for this article can be found online at http://dx.doi.org/10.7717/peerj.17116#supplemental-information.

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
