# Peer review of "Gene expression analysis of potato drought-responsive genes under drought stress in potato (Solanum tuberosum L.) cultivars"

_PeerJ, doi:10.7717/peerj.17116_

## Round 0.1 · original submission · Major Revisions

Dear Authors
The manuscript cannot be accepted for publication in its current form. It needs a major revision to be reconsidered for publication. The authors are invited to revise the paper considering all the suggestions made by all the reviewers, including those who reject the manuscript. Please note that requested changes are required for publication.

Additional comments:
1- The title of the manuscript must be rewritten and delete (twelve potatoes ).
2- all the gene's names must be written in italic
6- Besides, there are significant concerns about the manuscript's grammar, usage, and overall readability. Therefore, revise the text to fix grammatical errors and improve the text's overall readability. We suggest you have a fluent English-language speaker thoroughly copyedit your manuscript for language usage, spelling, and grammar. If you do not know anyone who can do this, PeerJ can provide language editing services.

With Thanks

**Language Note:** The Academic Editor has identified that the English language must be improved. PeerJ can provide language editing services - please contact us at copyediting@peerj.com for pricing (be sure to provide your manuscript number and title). Alternatively, you should make your own arrangements to improve the language quality and provide details in your response letter. – PeerJ Staff

Reviewer 1 ·

Basic reporting

in this study the authors investigates the activities of 12 different potato genes under different irrigation stresses in 4 different potato cultivars

Experimental design

the experimental was designed very well and according to the stander expermantes design

Validity of the findings

All the results are validated

Additional comments

The introduction section
This part was written will but you must add a paragraph to descript the importance of this genes for drought stress and mentioned the different related studies

Materials and Methods
1- In the part
In line 129 RNA extraction and Complementary DNA (cDNA) synthesis
You must start from the next line.
2- Quantitative Reverse Transcription PCR Data Analysis (RT-qPCR)
In line 145 (12 loci, listed in Table 1, were synthesized as primers. Real-time PCR was conducted using Real-Time PCR Master Mix (A.B.T.™ 2X qPCR SYBR-Green MasterMix (with ROX).
The paragraph must not begin with a number
You can use (A number of 12 loci ……………………..
3- In line 160 you write that you used one way ANOVA but I didn’t find any Table for this ANOVA if you mean that you explain the ANOVA using the figures (1,2,3,4,….) you must omit this sentences of ANOVA and write
The graphs of gene expression of STAREB2, STDHN1, ………. genes in all potato cultivars were done using GraphPad Prism version 8.0.0 (Motulsky, 2023) program
4- In line 162 (* Shows that the differences between means are significant (P<0.05), and ** Shows that the differences between means are highly significant statistically (P<0.01).) use this as legend under the figure.

5- In line 289 , 295 and 426 SUCROSE SYNTHASE writes it in lower case Sucrose synthase

6- Line 306 A1 (CONTROL) use control

7- The references needs to be revised

Annotated reviews are not available for download in order to protect the identity of reviewers who chose to remain anonymous.

Reviewer 2 ·

Basic reporting

In this manuscript, the lone author Celik attempts to study the gene expression pattern in different potato cultivars under water scarcity using qRT-PCR. In my opinion, the data presented in the manuscript is preliminary with no scientific validation therefore interpretation and inferences made by the author is speculative. Throughout the manuscript, authors employed qRT-PCR to show the expression at mRNA level, but authors did not validate the expression at protein level, which is necessary to make a conclusion, given that qRT-PCR itself has many pitfalls. Moreover, the manuscript is written poorly, and the flow of information could have been better. In addition to this, the manuscript lacks a clear hypothesis and scientific merits. English language should be improved as I found a lot of typos, grammatical errors and misrepresentation of scientific names.

Experimental design

See above

Validity of the findings

See above

Reviewer 3 ·

Basic reporting

The manuscript is interesting.

Experimental design

why you use the Dunnetts test for statistical analysis?

Validity of the findings

Please describe statistical significance in all figure's captions.

Additional comments

Please replace the old references with new ones.

---

## Round 0.2 · Major Revisions

Dear Authors

The manuscript cannot be accepted for publication in its current form. It needs a major revision to be reconsidered for publication. The authors are invited to revise the paper considering ALL the suggestions made by the reviewers. Please note that requested changes are required for publication.

With Thanks

Reviewer 3 ·

Basic reporting

Accept

Experimental design

Accept

Validity of the findings

Accept

Additional comments

Accept

Reviewer 4 ·

Basic reporting

In the study, the author examined the expression of 12 (or ten) genes that he stated were associated with drought in four potato cultivars under four different irrigation regimes using the qRT-PCR method. The manuscript has been previously reviewed by other reviewers and the author has attempted to make appropriate corrections. However, I reviewed it as if it were a newly submitted manuscript. The manuscript needs major revision. My comments are below.
Abstract
The abstract should be rewritten to be more descriptive. First of all, when I read the abstract, I had no idea what the author did or what the results he found would be useful.
Line 17: there are four different irrigation levels. It should be expressed as "control and three different drought treatments" or "four different irrigation levels".
Line 22: Did you find the same result at all drought levels? State this clearly. Classify stress as slightly drought, moderately drought, and severe drought and express it accordingly. Do this throughout the entire text.
Introduction
The first two sentences in the introduction can be combined. Avoid giving unnecessary information. Our topic is potatoes. Is there any need to introduce the Solanaceae family?
Line 43-45: What is the relevance of the chromosome number information of potatoes? You started talking about drought. You should change the order of the sentences. Maybe these can be used after the first sentence.
Line 69-70: Again, an irrelevant sentence is used. I think it should be deleted completely. It seems unnecessary.
Line 96: Oryza sativa should be written in italics. Correct the entire text, paying attention to the spelling of the species names of the plants.
Line 100-110: Our topic is genes expressed during drought stress. So, what is the status of the SUCROSE SYNTHASE (SuSy) gene during drought stress? You only mentioned its effects on yield. Also refer to its change during drought.
Line 111: Which 12 genes were studied should be written in parentheses.

Experimental design

Material and Method

I think it is not right to evaluate each potatoes variety separately in statistical analysis. The data should be analyzed to show the variety x drought interaction, and the graphs should be corrected and reinterpreted accordingly. Maybe this will also improve the appearance of the graphics. Graphs can be presented grouped by genes.

Line 131-134: You explained irrigation practices very complicatedly. "There were four different irrigation levels. It is much easier if you write it as A1: Control-100% field capacity, A2: slightly drought-75% field capacity, A3: moderately drought-50% field capacity, and severe drought-25% field capacity." understandable.
Line 187: correct spelling of 2-ΔΔCt
Line 189-191: You said initially it was examined 12 genes, but here you listed 10 gene names. Correct errors or complete deficiencies.

Validity of the findings

Results
Line 196-199: As you begin to present your results, is it necessary to say what you examined again? Also, why do you write all capital letters when writing gene and cultivar names? Please correct it so that only the first letter is capitalized throughout all text. If you used any abbreviation for them at the beginning of the text, just use that abbreviation in the following lines.
Line 200: Graphics need correction.
Line 202-203: Are you sure what you're describing here is compatible with the Figure 1? I think you didn't explain exactly what you wanted to say. In the cultivar you mentioned here, the expression of the gene you mentioned increased 7.5 times compared to the control in A2, and 2 times compared to the control in A3. But from what you said, it seems that A2 is 7.5 times more than the control and 2 times more than A3. But A2 is 3.75 times more than A3.
Line 222-224: Is this sentence really appropriate here? Follow an order when describing your results. You should explain it in the order in Figure 1. This way, it becomes difficult to keep track. However, when statistical calculations and graphics change, the results will change altogether. Pay attention to these when rewriting.
Discussion
Line 331-334: I don't think it's necessary to say what he did when he started explaining each section. Don't write these over and over again.
Line 334-335: Aren't you too late to point this out? I read about the 10 genes throughout the text. I wonder what the other two were.
Line 349: Are you sure?
Line 359: Is this MYB TFs? If it is not, This sentences is unnecessary.
The discussion section is insufficient. It should be improved and the number of references to recent literature should be increased. Very old ones can be deleted. The influence of genes should be better explained. Do not explain it this way unless there is clear evidence that protein is formed. You didn't validate them by looking at protein expression. Avoid very precise statements.
The conclusion section should be rewritten. Write down what you find and your suggestions without repeating the same sentences over and over.

Additional comments

no comment

Reviewer 5 ·

Basic reporting

no comment

Experimental design

no comment

Validity of the findings

no comment

Additional comments

See attached PDF

Annotated reviews are not available for download in order to protect the identity of reviewers who chose to remain anonymous.

Reviewer 6 ·

Basic reporting

1. The author used qRT-PCR to display the expression at the mRNA level, but the author did not show any information at the protein level.

This means the level of the mRNA of a gene could sometimes not reflect the protein level for the same gene, due to the posttranscriptional modification for example.

2. The manuscript has a critical issue with the sampling, timing, and quantification of the gene expression. So, I recommend rejecting the manuscript.

Experimental design

Not correct.
- The experimental design and gene expression analysis.

Validity of the findings

Primary results, the results are invalid.

Annotated reviews are not available for download in order to protect the identity of reviewers who chose to remain anonymous.

---

## Round 0.3 · Major Revisions

Dear Author
The reviewers have recommended revisions to your manuscript. Therefore, I invite you to respond to the reviewers' comments and revise your manuscript.
Please consider that the statistical analyses have a problem and need to be resolved, as the reviewer suggested. In addition, there are significant concerns about the manuscript's grammar, usage, and overall readability. We, therefore, request that you revise the text to fix the grammatical errors and improve the overall readability of the text. We suggest you have a fluent English-language speaker thoroughly copyedit your manuscript for language usage, spelling, and grammar. If you do not know anyone who can do this, PeerJ can provide language editing services.
With Thanks


**Language Note:** The Academic Editor has identified that the English language must be improved. PeerJ can provide language editing services - please contact us at copyediting@peerj.com for pricing (be sure to provide your manuscript number and title). Alternatively, you should make your own arrangements to improve the language quality and provide details in your response letter. – PeerJ Staff

Reviewer 4 ·

Basic reporting

The author took the comments into account to improve the manuscript. However, he did not change the statistical analysis section and graphical representation, stating that he did not understand the change I wanted. I explained it again. I hope it was understandable.

This is not the case when citations are given in the text, which must be placed in parentheses. Examine it carefully and correct the lines where parentheses need to be added.
Line 90-96: What is the status of Sucrose Synthase (SuSy) genes under drought? There is no information about this. Is the information you gave normal conditions for potatoes? In drought? In another stress situation? Why did you choose this gene? Why did you want to know Susy situation in the drought? If you don't connect this information to your purpose, it doesn't matter if you write it down.

Experimental design

Line 167: There should be a two-way ANOVA. In your response letter, you stated that you did not understand the problem with the graphics. You should also show the "cultivar x stress" interaction on the graph by doing a two-way ANOVA. You should use Cluster bar chart, not Simple. I think this statistic is wrong. Random blocks do not explain the differences between cultivars well. Interactions must be shown. When you do this, there is a single chart for each gene. You should draw a graph by placing the cultivars in the category axis and the drought levels in the clusters. Doing this will not change your gene expression levels. It does not affect the results. Only your analysis of variance comments changes and your graphical presentation improves.

Validity of the findings

Line 183: It should be “Results”. Not “Rresults”

The author should change the statistics and correct comments where necessary and resubmit the manuscript.

Additional comments

I'm not sure about the necessity of Figure 6.

Reviewer 5 ·

Basic reporting

no comment

Experimental design

no comment

Validity of the findings

no comment

Additional comments

no comment

Annotated reviews are not available for download in order to protect the identity of reviewers who chose to remain anonymous.

---

## Round 0.4 · Minor Revisions

Dear Authors
The manuscript still needs a minor revision to be reconsidered for publication. The author is invited to revise the paper considering all the suggestions made by the reviewer. Please note that the requested changes are required for publication.
With Thanks

Reviewer 4 ·

Basic reporting

The author made the requested corrections.

Experimental design

The author made the requested corrections.

Validity of the findings

The author made the requested corrections.

Additional comments

The author made the requested corrections.

Reviewer 5 ·

Basic reporting

no comment

Experimental design

no comment

Validity of the findings

no comment

Additional comments

See the attached file

Annotated reviews are not available for download in order to protect the identity of reviewers who chose to remain anonymous.

---

## Round 0.5 · accepted · Accept

Dear Authors,

I am pleased to inform you that after the last round of revision, the manuscript has been improved a lot, and it can be accepted for publication.

Congratulations on accepting your manuscript, and thank you for your interest in submitting your work to PeerJ.

With Thanks